# A New Case of Autosomal-Dominant *POLR3B*-Related Disorder: Widening Genotypic and Phenotypic *Spectrum*

**DOI:** 10.3390/brainsci13111567

**Published:** 2023-11-08

**Authors:** Vito Luigi Colona, Enrico Bertini, Maria Cristina Digilio, Adele D’Amico, Antonio Novelli, Stefano Pro, Elisa Pisaneschi, Francesco Nicita

**Affiliations:** 1Department of Biomedicine and Prevention, Tor Vergata University of Rome, 00133 Rome, Italy; vitoluigi.colona@opbg.net; 2Laboratory of Medical Genetics, Translational Cytogenomics Research Unit, Bambino Gesù Children’s Hospital, IRCCS, 00165 Rome, Italy; antonio.novelli@opbg.net (A.N.); elisa.pisaneschi@opbg.net (E.P.); 3Unit of Neuromuscular and Neurodegenerative Disease, Bambino Gesù Children’s Hospital, IRCCS, 00165 Rome, Italy; enricosilvio.bertini@opbg.net (E.B.); adele2.damico@opbg.net (A.D.); 4Genetics and Rare Disease Research Division, Bambino Gesù Children Hospital, IRCCS, 00165 Rome, Italy; mcristina.digilio@opbg.net; 5Medical Genetics Unit, Bambino Gesù Children Hospital, IRCCS, 00165 Rome, Italy; 6Developmental Neurology Unit, Bambino Gesù Children’s Hospital, IRCCS, 00165 Rome, Italy; stefano.pro@opbg.net

**Keywords:** ataxia, heterozygous, leukodystrophy, neuropathy, *POLR3A*, vertical gaze palsy, spinocerebellar

## Abstract

*POLR3B* encodes the RPC2 subunit of RNA polymerase III. Pathogenic variants are associated with biallelic hypomyelinating leukodystrophy belonging to the POLR-related disorders. Recently, the association with dominant demyelinating neuropathy, classified as Charcot–Marie–Tooth syndrome type 1I (CMT1I), has been reported as well. Here we report on an additional patient presenting with developmental delay and generalized epilepsy, followed by the onset of mild pyramidal and cerebellar signs, vertical gaze palsy and subclinical demyelinating polyneuropathy. A new heterozygous de novo missense variant, c.1297C > G, p.Arg433Gly, in *POLR3B* was disclosed via trio-exome sequencing. In silico analysis confirms the hypothesis on the variant pathogenicity. Our research broadens both the genotypic and phenotypic *spectrum* of the autosomal-dominant *POLR3B*-related condition.

## 1. Introduction

RNA polymerases (Pol) belongs to a family of complex holoenzymes, comprising numerous subunits, which catalyze the synthesis of copies of single-stranded RNA complementary to a template DNA strand (transcription). In humans, there are three RNA polymerases: Pol I transcribes rRNA genes; Pol II synthetizes mRNAs, snoRNAs, miRNAs, siRNAs, lncRNAs and most of the snRNAs; Pol III synthetizes tRNAs, 5S-rRNAs, some snRNAs (e.g., U6-snRNAs) and small cytoplasmic RNAs (scRNAs). Specificity is due to recognition of promotor sequences in proximity of the transcription start site [1].

*POLR3B* (MIM*614366) encodes Pol III Subunit B, the second largest subunit of Pol III (NX_Q9NW08) [1]. Biallelic variants in *POLR3B* are associated with a recessive condition, known as *POLR3B*-related or 4H-leukodystrophy (OMIM#614381), characterized by hypomyelination, cerebellar atrophy, with or without hypo-/oligo-dontia and hypogonadotropic hypogonadism, in the absence of nerve conduction abnormalities [2,3,4,5].

Additionally, de novo *POLR3B* heterozygous missense variants have been described in nine unrelated patients showing peripheral demyelinating neuropathy with or without central nervous system (CNS) involvement [6,7,8]. This form has been classified as demyelinating Charcot–Marie–Tooth syndrome type 1I (CMT1I, OMIM#619742). A dominant negative effect was postulated for all variants, potentially capable of altering the assembly of the functionally active polymerase III pre-initiation complex [9].

In this article, we report on a girl with developmental delay and early-onset drug-respondent generalized epilepsy, subsequently developing at her current age of 14 years a mild ataxic-spastic syndrome, vertical gaze palsy (VGP) and subclinical demyelinating neuropathy, correlated to a heterozygous genotype of an undescribed de novo variant in *POLR3B*.

## 2. Case Description

### 2.1. Clinical Report

A 14-year-old girl, the second child of non-consanguineous parents, comes to our attention for diagnostic evaluation because of a mixed developmental disorder with language delay, emotional-relational difficulties and epilepsy. The family history was uninformative. She was born at term from an uneventful delivery. Auxological parameters were in normal percentiles. The patient’s medical history included generalized myoclonic epilepsy with onset at age 16 months controlled by valproic acid, interrupted after a few years due to complete seizure control, and developmental delay with the acquisition of autonomous walking by the age of 2 years. An improvement in clinical course supported by psychomotor education and speech therapy, without signs of regression, was reported over years. A first 1.5T brain MRI disclosed a mild reduction in the cerebellar vermis at age 12 years (not shown). At age 14 years, an additional 3T brain MRI (Figure 1) confirmed a reduced volume of the cerebellar vermis and showed mild increase in inter-folia spaces in the superior vermis and in the cerebellar hemispheres without any evidence of cerebral or cerebellar white matter abnormalities.

On clinical examination at age 14, she presented a roundish face with non-specific dysmorphisms (i.e., wide eyebrows with partial sinophris, sunken orbital region, nose with bulbous tip and anteverted nostrils, arched upper lip), small hands, feet with partial cutaneous syndactyly between II-III-IV toes, substantially flat foot bilaterally, and postural kyphosis with scoliotic attitude with no clinical signs of polyneuropathy. Neurological examination disclosed inability to look upwards (i.e., VGP), hand stereotypies, mild ataxia appearing in tandem gait and subtle pyramidal signs (i.e., increased osteo-tendineous reflexes at four limbs, positive Hoffmann and Babinski sign). Motor nerve conduction velocities from tibial and peroneal nerves were 25.2 m/s and 22.5 m/s, respectively; F-wave from tibial nerve was absent, while sensory nerve conduction velocity from sural nerve was 37 m/s. These data suggested a demyelinating motor and sensory polyneuropathy at lower limbs. Motor-evoked potentials, recorded from tibialis anterior muscle, showed a bilateral increase in central motor conduction time (27.3 ms on the right side and 27.8 ms on the left side).

At last evaluation, she weighed 66 kg (75th percentile), she was 160.8 cm (50th percentile) tall (BMI = 25.5 kg/m^2^), and she had a head circumference of 51.5 cm (<3rd percentile). The Scale for the Assessment and Rating of Ataxia (SARA) was 5.5/40.

### 2.2. Genetics Results

After normal karyotype (46, XX) and SNP-array [arr(X,1-22)x2] results, a trio Clinical Exome Sequencing (CES) was performed. The latter highlighted the presence of a heterozygous de novo missense variant, NM_018082.6: c.1297C > G, p.Arg433Gly, in *POLR3B*, classified as likely pathogenic (class 4) according to the American College of Medical Genetics and Genomics (ACMG) criteria: PM1 (i.e., variant located in critical and well established functional domains), PM2 (i.e., absent from controls in population databases), PM5 (i.e., novel variant), PP2 (i.e., variant falling within a gene in which other missense variants represent a common pathogenetic mechanism), PP3 (i.e., multiple in silico evidence of a deleterious effect) and PS2 (i.e., de novo variant).

Through this investigation, further two novel heterozygous variants emerged, with maternal segregation, in two different distinct genes: a splicing variant (c.678 + 1G > A) in *ADAM22* (NM_004194.5); and a missense variant (c.707G > A, p.Arg236His) in *SMC3* (NM_005445.4). Both variants can be classified as class 4 according to the ACMG criteria: PSV1, PM2 for the variant in *ADAM22*; PM5, PP5, PM2 and PP2 for the one in *SMC3*. Finally, no relevant variants of uncertain significance (VUS, class 3) matching our patient’s clinical phenotype were detected in other genes.

### 2.3. Methods

Karyotype analysis was performed on metaphases from lymphocyte cultures according to standard G-banding techniques.

SNP-array analysis was performed using an Illumina BeadChip 850K platform (Illumina, San Diego, CA, USA), according to the manufacturer’s instructions. Images were analyzed using Bluefuse Multi software edition 4.5 (BlueGnome, Cambridge, UK).

CES analysis was performed using the Twist Custom Panels (Clinical Exome Twist Bioscience) according to the manufacturer’s protocol on the NovaSeq6000 platform (Illumina, San Diego, CA, USA). Reads were aligned to human genome build GRCh37/UCSC hg19. BaseSpace Dragen Enrichment application (Illumina) and TGex software (LifeMap Sciences, Alameda, CA, USA)—Geneyx Analysis v5.10 (knowledge-based NGS analysis tool powered by GeneCards Suite) were used for variant calling and annotation, respectively. The identified gene variants were assessed and screened using the TGex-Geneyx analysis software v5.10 to meet the following criteria: (1) non-synonymous exonic variants or intronic variants of ±5 bp; (2) minor allele frequency in the Genome Aggregation Database (GnomAD) less than 0.01 (1%); (3) call variant quality: coverage of ≥30X and GQ of ≥50; and (4) at least 20% of reads showing the alternative allele (Alt of >20%). Variants were visualized by Integrative Genomics Viewer. Sequence data were analyzed and the presence of all suspected variants was confirmed through examination of public databases (gnomAD, dbSNP, 1000 Genomes Project, EVS, ExAC). An in silico prediction of variant pathogenicity was achieved using Sorting Intolerant From Tolerant (SIFT), Polymorphism Phenotyping v2 (PolyPhen-2) and MutationTaster for the prediction of the single-nucleotide non-synonymous variant associated with human diseases. Variants were evaluated using VarSome and classified according to the ACMG criteria.

## 3. Discussion

### 3.1. In Silico Analysis

*POLR3B* (12q23.3) is composed of 27 introns and 28 exons, encoding for RPC2, a ubiquitously expressed protein of 1133 amino acids, which is one of the 17 subunits constituting the Pol III. The missense variant found in our patient is in exon 14 and is causative of a C > G transition at position 1297, leading to the substitution of an arginine (Arg) with a glycine (Gly) at position 433. Recently, eight missense monoallelic variants associated with a phenotype overlapping that of our case have been reported [6,7,8] (Figure 2A) (Appendix A). Some of these fall into an alpha-helix (p.Asp375Val; p.Ala365Val), or in a beta-sheet (p.Glu363Lys; p.Leu426Ser), between the lobe and the fork domains of RPC2, as observed in our patient. The p.Arg1046His, p.Thr462Arg and p.Arg469Cys variants fall into beta-sheets, respectively, in the clamp region (p.Arg1046His) [6,7,8] and within the fork domain, close to the catalytic site (p.Thr462Arg; p.Arg469Cys) [7,8]. Finally, only one variant, the p.Cys490Tyr, affects the hydrogen-bonded turn structure, in the fork domain (Appendix A) [8]. Given the dominant effect on the clinical phenotype, it is possible to state that the variants are in conserved sites [8].

Specifically, the c.1297C > G variant found in our patient reported a conservation score of 7 via ConSurf (Figure 2B) [10]. Arg433 is classified as a buried residue, which generally has greater tolerability to mutations compared to exposed active site residues, but from an analysis via SNAP2 [11], the p.Arg433Gly variant is predicted to impact the protein function (score 86, expected accuracy 91%) (Figure 2C). We can speculate that this predicted effect is due to the modification of the interaction network with the highly conserved (ConSurf score 8–9) proximal residues Gln183, Glu363, Asn423, Trp424, Met431 and Val436, caused by the replacement of a basic amino acid with a non-polar one (Figure 2D,E). This hypothesis is supported by the evaluation of the consequent change in protein stability via DynaMut2 [12], which identifies the substitution as destabilizing (ΔΔG^Stability^ = −0.83 kcal/mol). We expanded the in silico analysis through different phenotypic prediction platforms which returned the following results: “probably damaging” (Polyphen2 score 1); “pathogenic supporting” (SIFT score 0.01, G predicted not tolerated); “disease causing” (MutationTaster score 1). The CADD score is equal to 29.7, in support of the predicted pathogenicity of the variant found. The frequency of the variant is not known in gnomAD and was not found in parents (de novo). Finally, in ClinVar, a different change in the same aminoacidic position (p.Arg433Cys; ID: 1304259) is reported as being likely pathogenic.

### 3.2. Autosomal-Dominant POLR3B Phenotypic Spectrum

The autosomal-dominant *POLR3B*-related disorder presents phenotypically a *continuum* characterized mainly by a delay in motor development in the first years of life associated with progressive gait abnormalities, sensory ataxia, hyporeflexia and distal sensory impairment due to sensory-motor peripheral neuropathy affecting the lower limbs. A subgroup of patients also presents central nervous system involvement, which manifests as a global developmental delay with impaired intellectual and language abilities. Other features may include spasticity, hyperreflexia, tremor, dysmetria, ataxia and seizures. Brain imaging may be normal or show nonspecific abnormalities, in the absence of white matter impairment [6,7,8].

Clinically, our patient’s phenotype mostly overlaps with those reported in the other nine heterozygous *POLR3B* patients [6,7,8], which includes a delay in motor (*n* = 6/9, 67%) and language development (*n* = 4/9, 44%), severe (*n* = 2/9, 22%) and mild to moderate intellectual disability (*n* = 3/9, 33%), cerebellar and pyramidal signs (*n* = 5/9, 56%), epilepsy (*n* = 3/9, 33%), demyelinating peripheral neuropathy (*n* = 8/9, 89%) and neuroimaging features of cerebellar atrophy (*n* = 2/9, 22%) (Appendix A). Overall, isolated neuropathy has been reported in only four out of ten (40%) patients [6,7,8]. For this reason, in particular, clinicians should be aware that CNS involvement, likely originating from the ubiquitous expression of the Pol III, is common in this condition. Consequently, it is necessary to classify CMT1I not only as a pure inherited neuropathy, but as a complex disorder characterized by peripheral and additional signs of brain involvement.

A neurological examination of our patient also disclosed an inability to look upwards, consistent with VGP. Anomalies of vertical saccadic eyes movements (e.g., vertical gaze-evoked nystagmus, vertical saccadic paresis or palsy) are distinctive features mostly indicating damage in the midbrain, where centers for vertical eye functioning reside, but also in the *cerebellum* cerebellar (i.e., floccular lobe) [13]. The presence of VGP may underlie numerous causes such as acquired conditions (e.g., kernicterus, drugs), brain tumors or infections (e.g., Whipple’s disease) and several genetic conditions (e.g., Niemann–Pick syndrome type C, Tay–Sachs disease, Wilson’s disease, mitochondrial encephalopathies, primary dementias including tauopathies and alpha-synucleinopathies, Huntington’s disease, dentato-rubral-pallido-luysian atrophy, other neurodegenerative disorders including hereditary spinocerebellar ataxias, and neurodegeneration with iron accumulation) [13]. With respect to the genetically determined conditions, VGP represents a typical hallmark of very few diseases (e.g., Niemann–Pick Syndrome type C, childhood-onset neurodegeneration with ataxia, dystonia and gaze palsy due to *SQSTM1* biallelic variants) [14], while it can be randomly observed in all the above mentioned disorders, which are mostly characterized by predominant CNS rather than peripheral nervous system involvement. To the best of our knowledge, VGP has not been reported so far in autosomal-dominant *POLR3B*-related patients, but it has been reported in up to 20% of individuals affected by the autosomal-recessive *POLR3B* condition (i.e., 4H leukodystrophy) [15,16].

### 3.3. CES Secondary Molecular Findings

It was not possible to establish a significant association with the variants found in *ADAM22* and in *SMC3*. Moreover, we found only a single heterozygous variant c.678 + 1G > A in the *propositus* segregating from the healthy mother in *ADAM22* that is related to an autosomal recessive condition, nor was it possible to identify a link between our patient’s phenotype and the heterogeneous clinical manifestations associated with the *SMC3*-related dominant syndrome, whose characteristics are also not present in the clinically asymptomatic patient’s mother.

## 4. Conclusions

This article aims to corroborate the genotype–phenotype correlation of the autosomal-dominant *POLR3B*-related disorder for proper genetic counseling, and to expand the phenotype by introducing a new clinical sign, and expand the genotype by reporting a novel pathogenic variant. There are several limitations of this article: as for all single case reports, it cannot be used to draw definitive conclusions, which actually came from larger case series; additionally, the absence of functional studies does not allow us to go into the details of the variant pathogenicity. We are well aware that in silico analyses alone are not sufficient to support a defined causal role for the *POLR3B* variant. However, we are convinced that this approach can be considered a valid starting point for future investigations. Last but not least, CES analysis cannot identify the presence of variants in OMIM non-disease-causing genes, which are not analyzed via this method, and it also cannot detect the presence of deep intronic variants. Clinical and neuroimaging features in our patient together with a de novo heterozygous variant that can be classified as likely pathogenic support the diagnosis of an autosomal-dominant condition related to the *POLR3B*-related *spectrum*.

Monoallelic variants in *POLR3B* should be considered in the differential diagnosis of conditions [17] coupling developmental delay and peripheral neuropathy with or without spastic-ataxic features, thus suggesting that CMT1I does not represent a pure inherited neuropathy at least in a subgroup of cases. VGP may represent an additional sign to prompt the clinical diagnosis, but this needs to be confirmed in larger cohorts.

## Figures and Tables

**Figure 1 brainsci-13-01567-f001:**
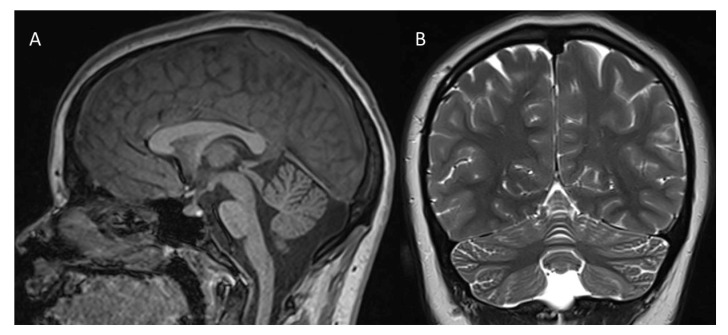
3T brain MRI ((**A**): sagittal T1-weighted image; (**B**): coronal T2-weighted image): cerebellar vermis appears of reduced volume (**A**,**B**); mild increase in the interfolial spaces can be noticed in the superior vermis and in the cerebellar hemispheres (**B**). No evidence of supra—(not shown) or sub-tentorial white matter involvement (**B**).

**Figure 2 brainsci-13-01567-f002:**
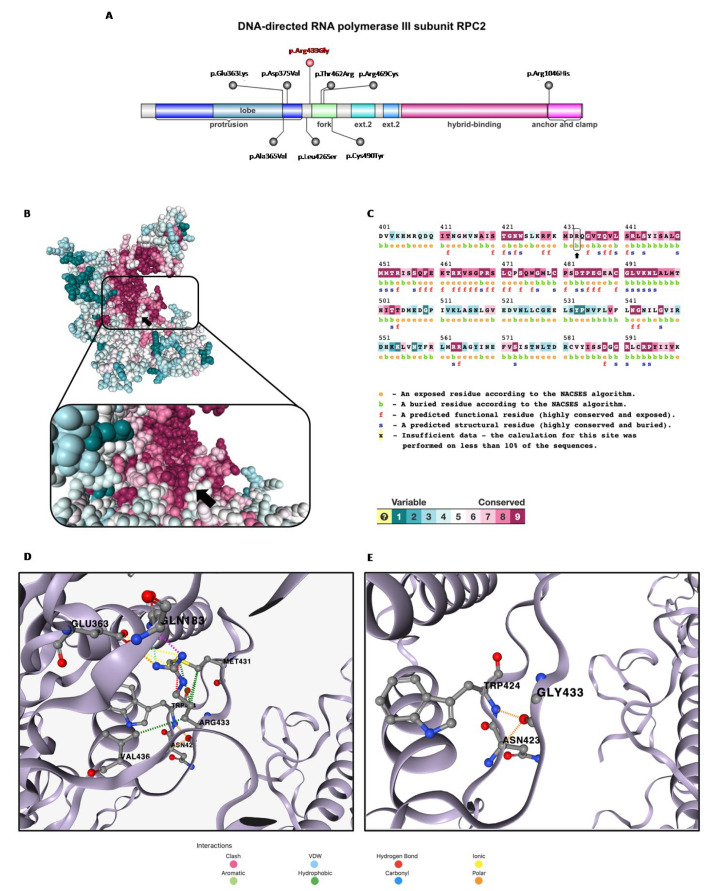
In silico structural prediction analysis: (**A**) domain representation of the DNA-directed RNA polymerase III subunit RPC2 with known *POLR3B*-dominant negative de novo missense variants (in red the one described in this report, Arg433Gly), generated by IBS 2.0. (**B**) Alphafold monomer v2.0 prediction for DNA-directed RNA polymerase III subunit RPC2 (ConSurf server). (**C**) RPC2 amino acid sequence generated by ConSurf server. The Arg433 residue is indicated with black arrow in both figures. (**D**,**E**) Disruption of the interaction network of Arg433 residue (wild-type) caused by substitution with Gly433 (mutant).

## Data Availability

Data sharing is not applicable to this article as no new data were created or analyzed in this study.

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
