# Peer review of "A New Case of Autosomal-Dominant *POLR3B*-Related Disorder: Widening Genotypic and Phenotypic *Spectrum"

_brainsci, 2023, doi:10.3390/brainsci13111567_

Round 1

Reviewer 1 Report (Previous Reviewer 1)

Comments and Suggestions for Authors

NO further comments.

Author Response

Thank you for your advice!

Reviewer 2 Report (Previous Reviewer 3)

Comments and Suggestions for Authors

The manuscript has been improved. I endorse publication in its current form.

Author Response

Thank you for your advice!

This manuscript is a resubmission of an earlier submission. The following is a list of the peer review reports and author responses from that submission.

Round 1

Reviewer 1 Report

Comments and Suggestions for Authors

Vito Luigi Colona, et al reported on A new case of autosomal-dominant POLR3B-related disorder: widening genotypic and phenotypic spectra.

The literature review and case are presented well. Minor comments include:

Substantial English language editing is required.

1.     Use “variant” instead of “mutation.”

2.     Use OMIM number, use NM_, NP_ number.

3.     Write the ACMG classification for the variant. How many were Class 1, class 2 and how many variants were VUS? VUS is not considered a confirmed diagnosis.

4.     Write proper variant nomenclature. [https://varnomen.hgvs.org/]

5.     Divide the results part into subheadings. It's confusing in this format.

Methods: mention the IRB approval number and details.

6. Was any control samples or database used to screen the variant? How many controls? Are they normal individuals or do they have some other disease?

6. In the last discussion add lines for future perspectives; discuss newborn screening, NIPT, PGT-A , PGT-M, for example.:

Proper genetic counseling for the affected family is essential in the case of rare genetic diseases. Furthermore, parenteral genetic screening/diagnosis is the best strategy for managing this disease, which currently has no therapy (PMID: 33613643). Reporting additional cases associated with this gene would help identify genotype–phenotype correlations and lead to clinical trials in the future?

**Any chance the most frequent variant could be added to the newborn screening??

**Any treatment given to the patients? Any medications? Improvements? 

Reviewer 2 Report

Comments and Suggestions for Authors

I suspect that one or two words may have gone astray around line 147.

Forty or fifty years ago, before genetic analysis was available, I suspect your case might have been regarded as a sporadic instance of olivo-ponto- cerebellar degeneration rather than of Charcot-Marie-Tooth disease - the shape of the floor of th 4th ventricle seems rather characteristic of the former, but it became increasingly recognised that the syndrome classification of instances of spino-cerebellar ataxia in an affected individual sometimes had to be altered as time passed after the initial designation and evidences of more widespread neurological disturbance became obvious. I wonder if this might happen in the case of your patient.

Reviewer 3 Report

Comments and Suggestions for Authors

Suggestions and comments for the Case Report  - A new case of autosomal-dominant POLR3B-related disorder: widening genotypic and phenotypic spectra by Vito Luigi Colona et al.

- This paper discusses a single case report, which lacks replication and control studies, the conclusions drawn appear over-ambitious given the data.

-The manuscript relies heavily on in silico analysis to make claims about pathogenicity.! these analyses are useful, they are not a substitute for functional studies to confirm the mutation's effects.

Also, the authors describe a clinical examination of a 14-year-old patient but fails to provide sufficient context about her history, precluding a comparative analysis. How does her case fit into the spectrum of previously described patients in the literature? Neurological signs are described, but the link between these signs and the POLR3B mutation is not thoroughly examined. This needs to be revised

They also suggest that their findings expand the phenotypic and genotypic spectrum of POLR3B-related disorders, yet the evidence provided seems insufficient to support such a broad claim.

The methods section does not adequately describe the technical and analytical approach information about the trio-exome sequencing is limited.

Regarding ACMG Guidelines - the variant is classified as 'likely pathogenic' according to the ACMG guidelines. This could be a strong point if adequately explained and substantiated, but the paper lacks detail here.

There is also an inadequate review of existing literature - how does this new case compare with the nine unrelated patients mentioned in the manuscript by authors?

Figure legends and their relevance to the text are not clearly outlined. This creates a disconnection between the data presented and the narrative.

The discussion is incomplete and does not adequately address the limitations of this manuscript.

 Consider revising accordingly.

Comments on the Quality of English Language

There are instances of poor sentence structure, which impair comprehension.